# Curcumin Reduces Depression in Obese Patients with Type 2 Diabetes: A Randomized Controlled Trial

**DOI:** 10.3390/nu16152414

**Published:** 2024-07-25

**Authors:** Metha Yaikwawong, Laddawan Jansarikit, Siwanon Jirawatnotai, Somlak Chuengsamarn

**Affiliations:** 1Department of Pharmacology, Faculty of Medicine Siriraj Hospital, Mahidol University, Bangkok 10700, Thailand; metha.yai@mahidol.ac.th (M.Y.); laddawan.jas@mahidol.ac.th (L.J.); siwawnon.jir@mahidol.ac.th (S.J.); 2Siriraj Center of Research Excellence for Precision Medicine and Systems Pharmacology, Faculty of Medicine Siriraj Hospital, Mahidol University, Bangkok 10700, Thailand; 3Faculty of Pharmacy, Silpakorn University, Nakhon Pathom 73000, Thailand; 4Division of Endocrinology and Metabolism, Faculty of Medicine, HRH Princess Maha Chakri Sirindhorn Medical Center, Srinakharinwirot University, Nakhon Nayok 26120, Thailand

**Keywords:** type 2 diabetes, curcumin, depression, serotonin, obesity

## Abstract

Type 2 diabetes and depression co-occur in a bidirectional manner. Curcumin supplements exhibit antidepressant effects that may mitigate depression by modulating neurotransmitters and reducing inflammatory and oxidative stress pathways. This study aimed to evaluate the efficacy of curcumin in improving depression severity in obese type 2 diabetes patients. The study employed a randomized, double-blind, placebo-controlled trial design with 227 participants. The primary end-point was depression severity assessed using the Patient Health Questionnaire-9. Biomarkers were measured at baseline and at 3-, 6-, 9-, and 12-month intervals. The biomarkers assessed were serotonin levels, pro-inflammatory cytokines (interleukin-1 beta, interleukin-6, tumor necrosis factor-alpha), antioxidant activities (total antioxidant status, glutathione peroxidase, and superoxide dismutase), and malondialdehyde. After 12 months, the curcumin group exhibited significantly improved depression severity (*p* = 0.000001). The curcumin group had higher levels of serotonin (*p* < 0.0001) but lower levels of interleukin-1 beta, interleukin-6, and tumor necrosis factor-alpha (*p* < 0.001 for all) than the placebo group. Total antioxidant status, glutathione peroxidase activity, and superoxide dismutase activity were elevated in the curcumin group, whereas malondialdehyde levels were greater in the placebo group (*p* < 0.001 for all). These findings suggest curcumin may have antidepressant effects on obese type 2 diabetes patients.

## 1. Introduction

Type 2 diabetes mellitus (T2DM) is a chronic condition characterized by elevated blood glucose levels due to dysfunction in β-cell biology and impaired insulin function. It is recognized as a significant global public health issue [1,2]. The worldwide prevalence of T2DM is rising, particularly in developed regions such as Western Europe, where the incidence is increasing rapidly. This condition affects both sexes equally, with the highest incidence typically occurring around the age of 55. Projections estimate that by 2030, the global prevalence of T2DM will reach 7079 individuals per 100,000, indicating a persistent upward trend globally [3]. Compared to their age-matched counterparts, T2DM patients are significantly more susceptible to disability, incapacity, and unemployment [4].

Major depressive disorder (MDD) is a clinical condition defined by a combination of at least five symptoms, where either depressed mood or loss of interest or pleasure must be present [5]. Additional symptoms include significant weight changes, insomnia or hypersomnia, psychomotor agitation or retardation, loss of energy, feelings of worthlessness or excessive guilt, diminished cognitive function, and recurrent thoughts of death [5]. Approximately 3.8% of the global population suffers from depression, with 5% of adults being affected. This number equates to approximately 280 million people worldwide [6].

Risk factors for MDD include female sex, middle age, unmarried status, low income, and disability. Moreover, a family history of depression, adverse childhood experiences, other mental health disorders, and chronic medical conditions such as T2DM increase the risk of developing MDD [5,7]. Clinically significant depression includes not only MDD but also subthreshold depression, which can lead to functional impairment and necessitate management. The prevalence of depression is significantly greater in individuals with T2DM than in the general population (19.1% vs. 10.7%) [8].

The co-occurrence of MDD and T2DM has cumulative effects. Affected individuals are more likely to experience disability-related work loss, noncompliance with medical treatment, and a greater risk of mortality than individuals with either condition alone [9]. MDD increases the risk of developing T2DM, and conversely, T2DM increases the risk of new or recurrent episodes of MDD, indicating a bidirectional relationship [10].

Antidepressants are the primary treatment for MDD but are associated with adverse effects, including impacts on cardiometabolic health and weight gain [11]. Studies suggest that pathways involving histamine and serotonin, which regulate appetite, contribute to these weight-related effects [12,13,14,15].

Curcumin is the primary curcuminoid found in turmeric, a rhizomatous root from the ginger family (Zingiberaceae). Known for its vibrant yellow color, turmeric is sometimes called “Indian saffron”. Its use dates back over 4000 years in southern Asia, where it is used both as a culinary spice and as a sacred component in religious rituals [16]. Curcumin has been shown to possess anti-inflammatory, antioxidant, and antiapoptotic properties [17,18,19]. The antidepressant effects of curcumin have been examined in various animal models of depression, such as the forced swimming test, tail suspension test, and chronic stress model [20,21,22]. These antidepressant effects are primarily due to two mechanisms: promoting neurogenesis in the hippocampus [23] and increasing the levels of serotonin, dopamine, and noradrenaline in the brain by inhibiting the monoamine oxidase enzyme [21,24].

Previous randomized clinical trials have indicated that curcumin, at doses ranging from 250 to 1000 mg per day over 10 to 12 weeks, may be beneficial for managing anxiety and depression, particularly in obese individuals and T2DM patients with major depression [25,26,27,28]. Despite these encouraging results, the number of randomized clinical trials investigating the effects of curcumin on depression in this population is limited, and existing studies have notable limitations. Therefore, the present study aimed to assess the efficacy of curcumin supplementation in improving depression severity among obese T2DM patients. We focused on increasing serotonin levels through the anti-inflammatory and antioxidant effects of curcumin. This evidence-based, double-blind, placebo-controlled clinical trial was designed to evaluate the feasibility of using curcumin as a therapeutic intervention.

## 2. Methods

### 2.1. Study Design and Participants

This randomized, double-blind, placebo-controlled trial was conducted at the HRH Princess Maha Chakri Sirindhorn Medical Center, Srinakharinwirot University, Thailand, with 260 T2DM patients. In a study lasting a total of 12 months, participants followed standardized diet and exercise protocols for an initial 3-month preparatory phase before being randomized for the 12-month main study period. Standard lifestyle recommendations, including medical nutrition therapy and physical activity guidelines, were provided and reinforced during a 20–30 min, one-on-one workshop.

The inclusion criterion was patients aged ≥35 years who were diagnosed with T2DM within the past 12 months. Patients were required to have a body mass index ≥ 23 kg/m^2^ and well-controlled blood glucose (glycated hemoglobin [HbA1c] < 6.5% and fasting plasma glucose [FPG] < 110 mg/dL). Diagnoses followed the American Diabetes Association 2017 guidelines [29]. The exclusion criteria were type 1 diabetes, impaired glucose tolerance, metabolic syndrome, maturity-onset diabetes of youth, gestational diabetes, uncontrolled hypertension, and dyslipidemia. Patients on antidiabetic drugs other than metformin were also excluded. None of the patients received insulin injection. Patients with hypertension and dyslipidemia were stably managed with antihypertensive and antidyslipidemic drugs, and no adjustments to these medication regimens were permitted during the study. Antihypertensive and antidyslipidemic drugs were represented in Appendix A. Blood samples were collected after overnight fasting at baseline and during visits at 0, 3, 6, 9, and 12 months. Patients with HbA1c ≥ 7.0% or FPG ≥ 130 mg/dL on two consecutive occasions during the intervention period were excluded. The flow of patients through the trial is summarized in the CONSORT diagram in Appendix A.

In order to assess dietary intake and exercise habits, the subjects completed a three-day food record and a dietary questionnaire at baseline and 12 weeks. The data were analyzed using Computer Dietary Guidance System Software (CDGSS 3.0; Appendix A).

The trial was approved by the Ethics Committee of the Faculty of Medicine at Srinakharinwirot University (approval number: SWUEC-176/58F) and registered with the Thai Clinical Trials Registry (TCTR20140303003). The study adhered to the Declaration of Helsinki, and informed consent was obtained from all participants.

### 2.2. Randomization Procedures

After screening, consent, and diet and lifestyle training, all participants were randomly assigned to either the curcumin-treated group (intervention) or the placebo-treated group (control). An independent researcher executed a fixed randomization scheme using computer-generated random numbers to determine group assignments. Allocation details were sealed in opaque, consecutively numbered envelopes, which the independent researcher opened sequentially. Participants were informed that two types of interventions were being compared.

### 2.3. Intervention Protocol

Participants were instructed to take three capsules of either curcumin or placebo twice daily, totaling six capsules per day, over 12 months. Each curcumin capsule contained 250 mg of curcuminoids. The Government Pharmaceutical Organization of Thailand manufactured the curcumin and placebo capsules. Compliance was monitored by asking participants to return all unused capsules during follow-up visits at 3, 6, 9, and 12 months, and capsule counts were recorded (Appendix A).

### 2.4. Preparation of Curcuminoid Capsules

Turmeric (*Curcuma longa* Linn.) rhizomes sourced from Kanchanaburi Province, Thailand, were dried and ground into a fine powder. The powder underwent ethanol extraction and low-pressure evaporation to produce a semisolid ethanol extract comprising oleoresin and curcuminoids. Subsequent oleoresin removal yielded a curcuminoid extract containing 75% to 85% total curcuminoids. High-performance thin-layer chromatography was used to quantify the peak ratios of curcumin, demethoxycurcumin, and bisdemethoxycurcumin in the extract. Following Good Manufacturing Procedures standards, the extract, standardized to contain 250 mg of curcuminoids, was encapsulated. Appendix A provides a fingerprint analysis of the extract and a detailed chemical composition review.

### 2.5. Study Outcomes

The primary outcome was assessed using the validated Thai version of the nine-item Patient Health Questionnaire (PHQ-9) [30,31]. The PHQ-9 scores ranged from 0 (no occurrence) to 3 (nearly every day), with total scores spanning from 0 to 27. Depression severity was categorized as follows: 0–4 (minimal), 5–9 (mild), 10–14 (moderate), 15–19 (moderately severe), and 20–27 (severe). The secondary outcomes included serum serotonin levels in order to assess the degree of depression [32,33]. Additionally, we measured pro-inflammatory cytokines (interleukin-1 beta [IL-1β], interleukin-6 [IL-6], and tumor necrosis factor-alpha [TNF-α]) and antioxidant activities (total antioxidant status, superoxide dismutase, and glutathione peroxidase). Adverse effects of curcumin were monitored through creatinine levels (≥1.2 mg/dL) and aspartate transaminase and alanine transaminase levels (≥3 times the upper limit of normal for either). Patient-reported symptoms were also recorded [34].

### 2.6. Data Collection and Measurement Methods

Measurements were conducted at baseline (pretreatment) and at 3, 6, 9, and 12 months postintervention. Baseline data included demographic information, medical history and medication questionnaire details, body weight, and height. Blood samples were drawn from the antecubital vein at 8:00 A.M. after an overnight fast, with patients in a recumbent position. Serum serotonin levels were quantified using a commercial enzymatic immunoassay kit (Serotonin ELISA Fast Track; Labor Diagnostika Nord, Nordhorn, Germany) and performed in duplicate per the manufacturer’s protocol. The analysis utilized a microplate reader set to 450 nm, with an enzyme-linked immunosorbent assay analytical range of 10.2 to 2500 ng/mL and a coefficient of variation of 9.7%. The normal reference range for serum serotonin was 70 to 270 ng/mL. Quality control samples at two concentrations were included in each assay.

Plasma samples for IL-1β, IL-6, and TNF-α assays were frozen and stored at −70 °C until analysis. All subjects were monitored for changes in cardiometabolic risk parameters, including FPG and HbA1c, over 1 year. These biomarkers were measured using standard procedures. Insulin resistance was evaluated using the Homeostatic Model Assessment of Insulin Resistance (HOMA-IR) [35]. The levels of the pro-inflammatory cytokines IL-1β, IL-6, and TNF-α were measured according to the manufacturer’s protocol (Abcam, Cambridge, UK). Serum total antioxidant status was determined using a novel automated method by Erel, which measures antioxidant capacity against hydroxyl radical reactions [36].

The serum activities of superoxide dismutase and glutathione peroxidase were analyzed colorimetrically using commercial kits (RANSOD and RANSEL kits; RANDOX Laboratory, Crumlin, UK). These analyses were performed on an automated analyzer (Alcyon 300; Abbott Laboratories, Abbott Park, IL, USA).

### 2.7. Sample Size

The sample size calculation was based on data from Chuengsamarn et al. [37] and employed a standard deviation of 160. A minimum of 113 subjects per treatment group were needed to detect a significant difference. After accounting for a 5% attrition rate, 269 subjects across two groups of similar size were selected to ensure adequate statistical power.

### 2.8. Statistical Analysis

Baseline demographic data are presented as the mean ± SEM for continuous variables and as counts and percentages for categorical variables. Continuous variables were compared using two-tailed Student’s *t* tests, and categorical variables were compared using chi-square tests, with significance set at *p* < 0.05. The outcome variables at 3, 6, and 9 months are reported as the mean ± SEM and were analyzed on an intention-to-treat basis. For within-group comparisons, paired samples *t* tests or Wilcoxon signed-rank tests were used, depending on the normality of the data. Between-group differences at 3, 6, 9, and 12 months were assessed using paired samples *t* tests or Wilcoxon signed-rank tests, as appropriate. Categorical variables were compared using Fisher’s exact test. All the statistical analyses were performed with R (version 4.1.2; R Foundation for Statistical Computing, Vienna, Austria), maintaining a significance threshold of *p* < 0.05.

## 3. Results

### 3.1. Participant Enrollment and Baseline Characteristics

Appendix A presents the CONSORT flowchart of the trial. Two hundred and sixty participants were initially enrolled and randomly assigned to the placebo or curcumin group. The baseline characteristics of the 227 subjects were comparable, with no significant differences between the two groups (Table 1).

### 3.2. Curcumin Treatment and Depression Severity

PHQ-9 scores, which assess depression severity, and serotonin levels were significantly lower in the curcumin group than in the placebo group at 3, 6, 9, and 12 months (Table 2). The curcumin group showed a marked improvement (20.4%) compared to the placebo group (2.63%; *p* < 0.000001; Table 3).

### 3.3. Glycemic Control Outcomes

The levels of diabetic indicators such as HbA1c and FPG were significantly lower in the curcumin group than in the placebo group at 6, 9, and 12 months (Table 2).

### 3.4. Insulin Resistance, Anti-Inflammatory, and Antioxidative Stress Outcomes

HOMA-IR, a clinical marker for insulin resistance, was analyzed in both the placebo and curcumin-treated cohorts. Compared with the placebo group, the curcumin group exhibited significantly reduced HOMA-IR levels at all follow-up intervals (3, 6, 9, and 12 months). Furthermore, the levels of the pro-inflammatory biomarkers IL-1β, IL-6, and TNF-α were significantly lower in the curcumin group than in the placebo group at the 6-, 9-, and 12-month visits (Table 2). The curcumin group also showed significant increases in total antioxidant status, glutathione peroxidase activity, and superoxide dismutase activity at the 3-, 6-, 9-, and 12-month intervals (Table 2). Conversely, malondialdehyde, an oxidative stress marker, was significantly lower in the curcumin group than in the placebo group at the 6-, 9-, and 12-month visits (Table 2).

### 3.5. Weight Measurement Results

The mean body weight and body mass index were significantly lower in the curcumin-treated group than in the placebo group at 3, 6, 9, and 12 months (Table 2).

### 3.6. The Effect of Curcumin between Genders

Comparison of means between genders shows no significant difference between males and females for any parameters such as PHQ-9, serotonin, and BMI (Appendix A).

### 3.7. Adverse Effects

The mild adverse effects were abdominal pain, diarrhea, and headache. None of the patients were dropped out due to adverse effects. To assess the potential adverse effects of curcumin, kidney and liver function tests were performed (Appendix A). No significant differences between the curcumin and placebo groups were found in aspartate transaminase, alanine transaminase, or creatinine. No hypoglycemia was observed in the curcumin group. Overall, these results suggest that curcumin extract can be safely used for at least 12 months. Capsule consumption was comparable between the groups, indicating similar compliance levels (Appendix A). Therefore, the observed effects were not due to differential compliance.

## 4. Discussion

Major depressive disorder is characterized by persistent sadness, anhedonia, suicidal ideation, and both somatic and cognitive symptoms. Individuals with MDD typically experience reduced quality of life due to the disorder, comorbid medical conditions, social challenges, and impaired daily functioning [38]. Depression enhances the risk of developing T2DM, which introduces complications such as hyperglycemia, insulin resistance, and vascular issues. Conversely, a T2DM diagnosis increases the risk and severity of depressive episodes due to shared etiological factors involving bidirectional interactions, such as autonomic and neurohormonal dysregulation, and inflammatory processes [39].

To identify a safe, well-tolerated, and accessible treatment to alleviate depression in T2DM patients, we evaluated ethanol-extracted curcumin. In our double-blind, placebo-controlled study, the subjects consumed 1500 mg/day of curcumin (turmeric root extract). Depression severity was assessed using the Thai version of the PHQ-9 questionnaire. Our findings indicate that curcumin consumption significantly reduced PHQ-9 scores, demonstrating a notable improvement over the placebo at the end of the 12-month treatment period (Table 3). Additionally, serotonin levels were significantly elevated in the curcumin-treated group.

Curcumin treatment has exhibited significant anti-inflammatory effects in in vivo models [40,41]. Our study revealed that a 6-month curcumin regimen significantly decreased IL-1β, IL-6, and TNF-α levels. The efficacy of curcumin in reducing the levels of these pro-inflammatory cytokines—IL-1β [42], IL-6 [43], and TNF-α [44]—is well documented. These cytokines impact serotonin synthesis, transport, metabolism, and receptor sensitivity, all of which are crucial factors in mood disorders such as depression [45,46]. Oxidative stress, defined by an imbalance between reactive oxygen species production and antioxidant defense, disrupts serotonin metabolism and neurotransmission. Reactive oxygen species directly impair serotonin synthesis and metabolism enzymes, including tryptophan hydroxylase and monoamine oxidase [47,48].

The interplay between oxidative stress and serotonin is implicated in various neurological and psychiatric conditions [49]. Curcumin has been demonstrated to reduce malondialdehyde levels and enhance the activities of antioxidant enzymes, such as superoxide dismutase, catalase, and glutathione peroxidase [50,51]. Our findings indicated increased antioxidant capacities, reflected by elevated total antioxidant status, glutathione peroxidase activity, and superoxide dismutase activity after 3 months of curcumin treatment. Conversely, malondialdehyde levels, which are indicative of lipid peroxidation and oxidative stress [52], were reduced following 6 months of curcumin administration in our investigation. Furthermore, curcumin treatment may have antidiabetic effects, as evidenced by our investigation’s reductions in FPG and HbA1c levels after 3 months of treatment.

The improvements in anti-inflammatory and antioxidant parameters likely contributed to the observed elevation in serotonin levels within our curcumin-treated group, leading to a reduction in depression severity. Conventional antidepressants, while effective, can induce weight gain by affecting neurotransmitters such as serotonin and histamine, which regulate appetite and metabolism [53,54]. In contrast, our study demonstrated a significant reduction in body weight and body mass index in the curcumin-treated group, potentially linked to enhanced insulin sensitivity, improved glycemic control, and decreased cardiometabolic risk factors [55]. Additionally, we observed a significant reduction in BMI in the curcumin-treated group (Table 2). The effect of curcumin on lowering BMI may vary [56,57,58]. However, our understanding of its effectiveness in achieving lower BMI levels is incomplete, and further research is necessary for a comprehensive understanding. It may be possible that in some cases, significant weight loss may have had a positive effect on the depression severity; however, we do not have any direct evidence to support it from our study.

Regarding safety, the oral administration of curcumin is well-documented as safe. Human studies suggest that curcumin can be tolerated at high doses of up to 8000 mg/day without evident toxicity [59]. This concurs with the findings of our investigation, which used a dose of 1500 mg/day without producing any severe side effects. Various randomized clinical trials have indicated that curcumin has the potential to ameliorate depression severity [25,26,27,28]. However, these studies were often limited by small sample sizes (30–80 participants), brief intervention periods (10–12 weeks), and a paucity of safety, inflammatory, and oxidative stress data.

Our study was specifically designed to address the limitations of previous investigations by enrolling a large cohort of 227 subjects and having a prolonged follow-up period of 12 months. The results demonstrated that curcumin extract effectively ameliorated depression severity in obese T2DM patients. Our findings suggest that the anti-inflammatory properties of curcumin might elevate serum serotonin levels. Specifically, pro-inflammatory cytokines such as TNF-α, IL-1β, and IL-6 can activate indoleamine 2,3-dioxygenase, an enzyme that degrades tryptophan—a precursor of serotonin [60].

The antioxidant effects of curcumin may also increase serum serotonin levels. Oxidative stress is a significant factor in depression development, disrupting the stress response, causing neuroinflammation, and altering serotonin levels [61]. The proposed curcumin effect on reducing depression was presented in Figure 1. However, our study has limitations. These include its single-dose design, which precludes analysis of potential dose–response relationships. Additionally, the fact that it is a single-center randomized controlled trial may limit the generalizability of the findings to other populations or settings.

## 5. Conclusions

Curcumin supplements exhibit potential antidepressant effects on type 2 diabetes patients with obesity by elevating serotonin levels, reducing inflammation, and mitigating oxidative stress. Our study demonstrated that curcumin may effectively alleviate depression severity in this population.

## Figures and Tables

**Figure 1 nutrients-16-02414-f001:**
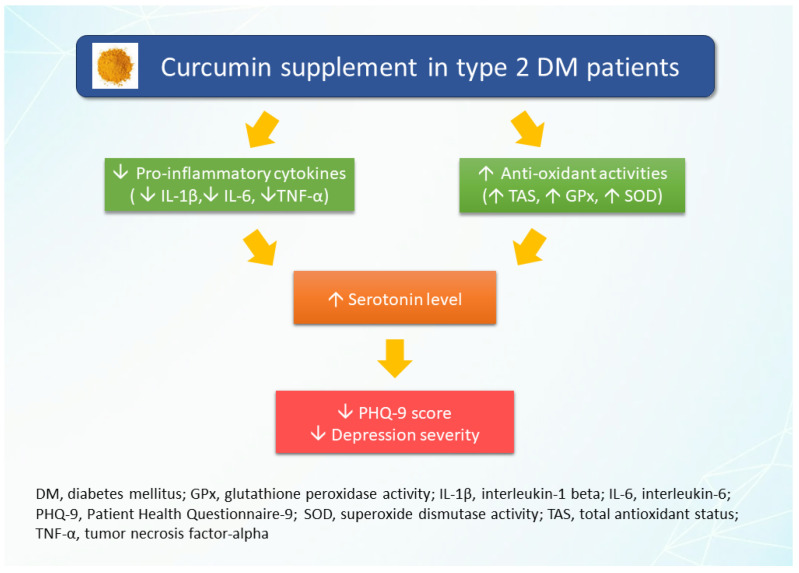
The proposed of curcumin effect on reducing depression.

**Table 1 nutrients-16-02414-t001:** Baseline Demographic and Clinical Characteristics of the Study Participants.

Variable	Placebo	Curcumin	*p* Value *
	Mean (SEM) (*n* = 114)	Mean (SEM) (*n* = 113)	
Sex, M:F ratio	54/80 (0.67)	62/73 (0.85)	0.87 ^†^
Age, y	62.26 (0.81)	60.27 (0.83)	0.13
BMI, kg/m^2^	26.76 (0.38)	27.21 (0.37)	0.41
Weight, kg	69.50 (1.32)	69.92 (1.24)	0.58
Systolic blood pressure	129.25 (1.28)	129.76 (1.30)	0.95
Diastolic blood pressure	75.84 (1.05)	75.14 (1.15)	0.95
PHQ-9	NA	NA	NA
Serotonin, ng/mL	NA	NA	NA
FBG, mg/dL	125.80 (2.22)	123.65 (1.73)	0.401
HbA1c, %	6.26 (0.06)	6.28 (0.07)	0.69
HOMA-IR, units	5.24 (0.24)	5.38 (0.23)	0.72
IL-1β, pg/mL	0.44 (0.02)	0.42 (0.02)	0.46
IL-6, pg/mL	8.71 (0.11)	8.96 (0.12)	0.34
TNF-α, pg/mL	5.01 (0.14)	4.78 (0.13)	0.24
TAS, μmol/trolox eq/L	1.60 (0.01)	1.58 (0.01)	0.30
Glutathione peroxidase activity, U/mL	6583.78 (218.65)	6693.82 (206.44)	0.90
Superoxide dismutase activity, U/mL	241.09 (4.60)	237.59 (4.38)	0.77
Malondialdehyde, μmol/L	2.01 (0.04)	1.95 (0.04)	0.28
Creatinine, mg/dL	0.87 (0.02)	0.86 (0.02)	0.77
AST, U/L	25.01 (0.87)	25.34 (0.80)	0.58
ALT, U/L	27.58 (1.56)	30.09 (1.50)	0.08
History of cerebrovascular disease	7 (6.1%)	5 (4.4%)	0.78 ^†^
History of coronary artery disease	9 (7.8%)	8 (7.1%)	1.00 ^†^
History of hypertension	82 (71.9%)	76 (67.2%)	0.53 ^†^
History of diabetic nephropathy	18 (15.8%)	28 (24.8%)	0.13 ^†^
History of dyslipidemia	104 (77.6%)	101 (74.8%)	0.84 ^†^

ALT, alanine transaminase; AST, aspartate aminotransferase; BMI, body mass index; FBG, fasting plasma glucose; HbA1c, glycated hemoglobin; HOMA-IR, homeostatic model assessment of insulin resistance; IL-1β, interleukin-1 beta; IL-6, interleukin-6; NA, not applicable; PHQ-9, Patient Health Questionnaire-9; TAS, total antioxidant status; TNF-α, tumor necrosis factor-alpha * All data except sex (M:F ratio) were evaluated by the Mann–Whitney U test. ^†^ Chi-square test.

**Table 2 nutrients-16-02414-t002:** PHQ-9 Score, Body Composition, and Chemistry Biomarker Measures.

Outcomes	Follow-Up Period (mo)	Placebo		Curcumin		*p* Value
		Mean	Minimum–Maximum	Mean	Minimum–Maximum	
PHQ-9	0	11.22	3–15	11.59	3.00–15.00	NS
	3	11.81	5–15	9.97	3.00–14.00	<0.0001
	6	12.23	5–15	8.91	3.00–14.00	<0.0001
	9	12.48	4–15	8.26	3.00–13.00	<0.0001
	12	12.84	6–16	7.66	2.00–13.00	<0.0001
Serotonin, ng/mL	0	99.51	70.40–132.00	100.39	71.28–132.00	NS
	3	103.26	70.40–132.00	104.23	70.40–132.00	NS
	6	102.44	48.14–154.35	136.50	87.35–198.76	0.0001
	9	101.23	48.19–153.56	143.35	94.35–193.33	<0.0001
	12	100.60	46.59–150.23	151.03	99.87–199.87	<0.0001
HbA1c, %	0	6.26	4.80–8.90	6.28	4.40–9.50	NS
	3	6.44	5.00–8.90	6.26	4.70–9.20	<0.01
	6	6.46	5.10–9.00	6.25	4.50–8.30	<0.01
	9	6.47	5.00–10.40	6.19	4.10–8.20	<0.05
	12	6.47	5.00–10.50	6.12	4.20–8.40	<0.05
Glucose	0	125.08	91–285	123.65	79–178	NS
	3	128.93	100–195	124.40	80–171	NS
	6	130.34	77–231	122.82	79–204	<0.01
	9	130.93	97–201	118.67	75–165	<0.01
	12	130.71	98–194	115.49	70–160	<0.05
HOMA-IR	0	5.24	1.70–21.80	5.38	1.20–14.20	NS
	3	5.88	2.00–17.00	5.25	1.70–12.80	<0.05
	6	5.93	1.80–17.90	5.17	1.60–16.50	<0.05
	9	6.02	2.20–19.80	5.02	1.30–11.50	<0.05
	12	6.04	2.30–18.00	4.86	1.20–11.00	<0.05
IL-1β, pg/mL	0	0.44	0.01–0.86	0.46	0.01–0.88	NS
	3	0.46	0.02–0.87	0.45	0.01–0.87	NS
	6	0.71	0.20–1.74	0.43	0.15–1.54	<0.001
	9	0.72	0.20–1.65	0.41	0.12–0.99	<0.001
	12	074	0.32–1.86	0.31	0.10–1.39	<0.001
IL-6, pg/mL	0	8.71	7.04–10.56	8.96	7.04–10.56	NS
	3	8.89	7.04–10.56	8.72	7.04–10.56	NS
	6	12.84	5.21–17.99	7.54	3.11–14.99	<0.001
	9	14.30	7.65–19.66	6.82	3.2–13.24	<0.001
	12	15.84	4.33–19.66	6.12	3.09–12.40	<0.001
TNF-α, pg/mL	0	5.01	2.64–7.04	4.77	2.64–7.04	NS
	3	5.16	2.64–7.04	4.84	2.64–7.04	NS
	6	5.91	2.18–14.88	4.23	1.46–10.5	<0.001
	9	6.37	2.24–14.98	3.81	1.43–9.44	<0.001
	12	6.77	2.14–15.37	3.46	1.33–8.59	<0.001
TAS, μmol trolox eq/L	0	1.60	1.20–1.98	1.59	1.25–1.89	NS
	3	1.61	1.26–2.20	1.70	1.35–2.14	<0.05
	6	1.63	1.20–1.98	1.73	1.32–2.01	<0.05
	9	1.63	1.09–2.94	1.82	1.42–2.94	<0.05
	12	1.63	1.21–2.45	1.86	1.49–2.58	<0.05
RANSEL, U/mL	0	6583.78	1083–13,199	6693.82	1124–13,012	NS
	3	6537.38	3540–19,484	7380.96	3497–15,378	<0.05
	6	6501.03	2500–18,386	8273.31	4376–17,679	<0.05
	9	5675.5	3090–10,834	11,048.48	6089–16,548	<0.001
	12	5153.41	3489–10,234	13,143.50	5787–20,987	<0.001
RANSOD, U/mL	0	241.09	133.00–379.00	241.59	144.00–362.00	NS
	3	237.19	133.00–420.00	244.93	151.00–362.00	<0.05
	6	224.35	120.00–420.00	265.74	144.00–450.00	<0.001
	9	202.45	120.00–280.00	280.38	189.00–450.00	<0.001
	12	176.34	112.00–218.00	328.90	215.00–468.00	<0.001
MDA, μmol/L	0	2.01	1.20–3.30	2.00	1.20–3.23	NS
	3	2.03	1.08–3.42	2.01	1.03–4.22	NS
	6	2.32	1.15–3.90	1.90	0.99–3.51	<0.001
	9	2.34	1.22–5.59	1.66	0.84–3.28	<0.001
	12	2.40	1.20–5.59	1.45	0.93–2.60	<0.001
BMI, kg/m^2^	0	26.94	16.45–35.18	27.35	20.40–36.58	NS
	3	26.98	16.88–40.37	26.56	19.15–44.81	<0.05
	6	26.96	17.31–40.79	25.90	18.31–43.71	<0.01
	9	28.00	16.88–40.79	26.02	19.14–42.61	<0.001
	12	29.34	17.72–42.11	25.94	17.90–42.24	<0.001
Body weight, kg	0	69.50	61–112	69.92	71–120	NS
	3	69.53	62–113	67.72	70–117	<0.05
	6	69.47	63–119	66.06	69–135	<0.01
	9	72.01	63–117	66.08	68–117	<0.01
	12	75.30	63–140	63.85	68–114	<0.001

HbA1c, glycated hemoglobin; HOMA-IR, homeostatic model assessment of insulin resistance; IL-1β, interleukin-1 beta; IL-6, interleukin-6; MDA, malondialdehyde; NS, not significant; PHQ-9, Patient Health Questionnaire-9; RANSEL, glutathione peroxidase activity; RANSOD, superoxide dismutase activity; TAS, total antioxidant status; TNF-α, tumor necrosis factor-alpha.

**Table 3 nutrients-16-02414-t003:** Comparison of Depression Severity Assessed by the Patient Health Questionnaire-9 Within and Between Study Groups.

Severity of Depression		Placebo (*n* = 114)			Curcumin (*n* = 113)			*p* Value *
		Baseline	12 Months	Improved ^ⵜ^	Baseline	12 Months	Improved ^ⵜ^	
PHQ-9	Minimal, (0–4)	3 (2.63%)	1 (0.88%)	3/114 (2.6%)	NA	9 (7.96%)	23/113 (20.4%)	<0.000001
	Mild, (5–9)	15 (13.16%)	8 (7.02%)		17 (15.05%)	91 (80.53%)		
	Moderate, (10–14)	93 (81.58%)	94 (82.45%)		89 (78.76%)	13 (11.51%)		
	Moderately Severe (15–19)	3 (2.63%)	11 (9.65%)		7 (6.19%)	NA		

NA, not applicable; PHQ-9, Patient Health Questionnaire-9 ^ⵜ^ Depression severity improvement by ≥1 stage. * Comparison of change values between the study groups.

## Data Availability

The original contributions presented in the study are included in the article, further inquiries can be directed to the corresponding author.

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
