# Peer review of "Curcumin Reduces Depression in Obese Patients with Type 2 Diabetes: A Randomized Controlled Trial"

_nutrients, 2024, doi:10.3390/nu16152414_

Round 1

Reviewer 1 Report

Comments and Suggestions for Authors

In the manuscript entitled ” Curcumin Reduces Depression in Obese Patients with Type 2 Diabetes: A Randomized Controlled Trial” the Authors tried to reveal the connection of curcumin supplementation in T2D patients on depression incidence. The study is very interesting for the readers since it is clinical and may reveal the important impact of curcumin on neurological diseases  as well as on emotional behavior. In my opinion the manuscript has been properly written, the methodology has been explained with the details, and the conclusions are connected with the obtained results. Frankly speaking, this is the first manuscript in which I do not have many recommendations for the Authors – and it is connected with really properly thought and described experiment treated as controlled trial. In my opinion, the study has been properly planned, and it demonstrates the real impact of long-term taken at least 1.5 g of curcumin on the metabolism parameters of diabetic subjects, such as reduction of malondialdehyde, FPG, HbA1c, IL-1β, IL-6, and TNF-α, whereas the increase of antioxidant molecules was observed (i.e., total antioxidant status, glutathione peroxidase activity, and superoxide dismutase activity, serotonin level). 

I am only very curious what was the impact of curcumin intake on the status of women and men, as well as on the similar parameters’ values in case of healthy human subjects (also divided on men and women). I would also suggest to try to present the correlation/effect of curcumin on T2D connected parameters. Have the Authors checked any impact of curcumin on the physical strength and willingness to play sports?

In summary, I would like to congratulate  the perfectly written manuscript and suggest its acceptance after inclusion of the additional data mentioned above (hoping that the Authors have still the access to these data). The Authors can also include a Figure as a scheme informing about the different molecular mechanisms regulated by curcumin. 

In summary, I suggest the minor revision, however I would gladly read the answer of Authors.

Author Response

Comment 1:
I am only very curious what was the impact of curcumin intake on the status of women and men, as well as on the similar parameters’ values in case of healthy human subjects (also divided on men and women).

Response:
We added section 3.6, the effect of curcumin between genders as highlighted (line 245-248).

“3.6. The effectof curcumin between genders

Comparison of means between genders shows no significant difference between males and

females for any parameters such as PHQ-9, serotonin, and BMI (Supplementary Table S4).”

Unfortunately, we did not collect any data about the effect of curcumin on healthy subjects.

Comment 2:

I would also suggest to try to present the correlation/effect of curcumin on T2D connected parameters.

Response:

Thank you for the suggestion. The effect of curcumin on type 2 diabetes were presented in result section 3.3. Glycemic Control Outcomes.

Comment 3:

Have the Authors checked any impact of curcumin on the physical strength and willingness to play sports?

Response:

Thank you for the interesting point. Unfortunately, we did not gather any data on the impact of physical strength and the willingness to engage in sports.

Comment 4:

The Authors can also include a Figure as a scheme informing about the different molecular mechanisms regulated by curcumin.

Response:

In the revised version, we added the sentence “The proposed of curcumin effect on reducing depression were present in figure 1.”as highlighted (line 330-331) and figure 1. The proposed of curcumin effect on reducing depression were added.

Reviewer 2 Report

Comments and Suggestions for Authors

1) This study is very interesting as it suggests that curcumin may reduce depression severity and have positive effects on glucose tolerance. However, I think the description of the side effects of curcumin is too limited. Did the changes in liver values differ between the two groups? Were there any patients who dropped out due to physical symptoms as side effects?

2) Are there any parameters related to the reduction of PHQ-9 scores? For example, are changes in inflammatory markers associated with the reduction of PHQ-9 scores?

3) The changes in BMI between the two groups are strikingly different. Can this difference really be explained by the effects of curcumin?

4) The font sizes of the text in the table are inconsistent, making it very difficult to read.

Author Response

Comment 1:

This study is very interesting as it suggests that curcumin may reduce depression severity and have positive effects on glucose tolerance. However, I think the description of the side effects of curcumin is too limited. Did the changes in liver values differ between the two groups? Were there any patients who dropped out due to physical symptoms as side effects?

Response:

Thank you very much for the encouraging comment. In the revised version, we added a new section 3.7. Adverse effects as highlighted (line 249-258).

“3.7. Adverse effects

The mild adverse effects were abdominal pain, diarrhea, and headache. None of the patients were dropped out due to adverse effects. To assess the potential adverse effects of curcumin, kidney and liver function tests were performed (supplementary table S5). No significant differences between the curcumin and placebo groups were found in aspartate transaminase, alanine transaminase, or creatinine. No hypoglycemia was observed in the curcumin group. Overall, these results suggest that curcumin extract can be safely used for at least 12 months. Capsule consumption was comparable between the groups, indicating similar compliance levels (supplementary table S3). Therefore, the observed effects were not due to differential compliance.”

Were there any patients who dropped out due to physical symptoms as side effects?

In section 3.7. Adverse effects as highlighted, we added the sentence “None of the patients were dropped out due to adverse effects.” (line 250-251).

Comment 2:

Are there any parameters related to the reduction of PHQ-9 scores? For example, are changes in inflammatory markers associated with the reduction of PHQ-9 scores?

Response:

We did not find any parameters (pro-inflammatory cytokines, antioxidants activities, and malondialdehyde) were associated with the reduction of PHQ-9 scores.

Comment 3:

The changes in BMI between the two groups are strikingly different. Can this difference really be explained by the effects of curcumin?

Response:

Thank you very much. We observed the BMI reduction as pointed out by the reviewer. However, further work is still needed to explain the reduction. It is possible that reduction of BMI may have had a positive effect on the depression severity. However, we still do not have any direct evidence to support it.

In the revised version, we discussed the reduction of BMI in the treatment group as highlighted (line 306-312).

“Additionally, we observed a significant reduction in BMI in the curcumin-treated group (Table 2). The effect of curcumin on lowering BMI may vary 56-58. However, our understanding of its effectiveness in achieving lower BMI levels is incomplete, and further research is necessary for a comprehensive understanding. It may be possible that in some case significant weight loss may have had a positive effect on the depression severity, however we do not have any direct evidence to support it from our study.”

Comment 4:

The font sizes of the text in the table are inconsistent, making it very difficult to read.

Response:

We adjusted the font sizes of the text in the table to the same font size (size 10).

Round 2

Reviewer 2 Report

Comments and Suggestions for Authors

no more comments to be left